# The Impact of Swallowing Difficulties on Quality of Life in Adults with Intellectual Disabilities in Residential Care: Cross-Sectional Study

**DOI:** 10.3390/ijerph22101470

**Published:** 2025-09-24

**Authors:** Maya Izumi, Seijun Ganaha, Yuki Kai, Ikuri Konishi, Risa Kira, Sumio Akifusa

**Affiliations:** 1School of Oral Health Sciences, Faculty of Dentistry, Kyushu Dental University, Fukuoka 803-8580, Japan; 2Kumamoto Kizuna Dental Office, Kumamoto 861-5255, Japan; 3Aso Kizuna Dental Office, Kumamoto 869-2612, Japan

**Keywords:** intellectual disability, swallowing function, health-related quality of life

## Abstract

Dysphagia is a common but often overlooked concern among individuals with intellectual disabilities and may negatively impact their quality of life. This study aimed to examine the association between swallowing function and health-related quality of life (HRQOL) among adults with intellectual disabilities in residential care. Methods: We conducted a cross-sectional study involving 48 individuals in residential care facilities in Japan. Swallowing function was assessed with the Eating Assessment Tool-10 (EAT-10), and HRQOL with the EuroQol-5 Dimension, five-level version (EQ-5D-5L), in which higher index values denote better HRQOL. A generalized linear model (GLM) with gamma distribution was used to identify factors associated with EQ-5D-5L scores. Participants with lower EAT-10 scores (≤2) showed significantly higher EQ-5D-5L scores. In the GLM, better swallowing function was positively associated with EQ-5D-5L score (B = 0.21, *p* = 0.012). Sensitivity analysis confirmed the robustness of this association. Conclusions: Swallowing dysfunction is linked to reduced HRQOL in individuals with intellectual disabilities, underscoring the need for regular screening and intervention.

## 1. Introduction

Dysphagia is a critical health concern among individuals with intellectual disabilities, not only due to its prevalence, but also because of its serious consequences. One of the most significant complications is aspiration pneumonia, which has been identified as a leading cause of morbidity and mortality in this population [1,2]. In addition to pneumonia, dysphagia has been associated with other life-threatening risks, such as choking and airway obstruction [3,4], as well as with deteriorations in nutritional status and hydration, which can further compromise overall health and well-being [5]. Despite this, the potential impact of dysphagia on the quality of life (QOL) of individuals with intellectual disabilities, including conditions such as Down syndrome and intellectual disability co-occurring with autism spectrum disorder, has received limited empirical attention [6]. While clinical management of dysphagia has received considerable attention, evidence linking swallowing function to health-related quality of life (HRQOL) in adults with intellectual disabilities remains limited. To address this gap, we examined the association between EAT-10 scores and EQ-5D-5L utilities in residents of long-term care facilities. In addition, communication difficulties in people with learning disabilities can hinder the safe management of eating and drinking; augmentative and alternative communication (AAC) may therefore be required and should be considered in clinical planning [7]. Population-level data in adults with intellectual disabilities are scarce: the few near-representative studies estimate dysphagia at ~8–12% [8]. Behavioral predictors of asphyxiation/choking include speed of eating, cramming food, and premature pharyngeal bolus loss [3]. Natural-history data show markedly elevated downstream risk: adults with intellectual disability are more likely to be hospitalized for, or die from, aspiration pneumonia [9,10], with standardized mortality up to ~35-fold higher than the general population [11].

Numerous studies have explored the QOL of individuals with intellectual disabilities. For example, in children with intellectual disabilities, pain, night-time sleep disturbances, daytime somnolence, seizures occurring at least weekly, and conservatively managed severe scoliosis have been reported to negatively affect QOL [12]. While self-determination has been identified as a key contributor to QOL among individuals with intellectual disabilities [13]. Cross-cultural work comparing Japan and Denmark suggests that domains such as personal development and self-determination are less emphasized in Japan, implying cultural influences on QOL [14]. Against this background, the EuroQol-5 Dimension, five-level version (EQ-5D-5L), provides a standardized, preference-based measure of health-related QOL (HRQoL) with country-specific value sets; the same health state can yield different index values across countries, supporting culturally sensitive interpretation while cautioning against direct between-country comparisons.

Impaired swallowing is associated with lower HRQoL [15]. Yet, the public health literature has largely emphasized older adults in general, with limited evidence specific to intellectual disabilities, particularly on how dysphagia relates to HRQoL when assessed with standardized measures. This study addresses this gap by quantifying the association between swallowing difficulties and HRQoL in adults with intellectual disabilities, generating evidence to inform screening, prevention, and service planning for this underserved population.

We hypothesized that reduced swallowing function would be associated with lower HRQOL among adults with ID. Because dysphagia likely intersects bidirectionally with nutritional status, functional independence, oral function (including oral bacterial burden), and body composition (e.g., via reduced intake, deconditioning, and sarcopenia), and because evidence integrating these domains among individuals with intellectual disabilities is sparse, we pre-specified these variables for measurement and analytical adjustment to contextualize EAT-10 and EQ-5D-5L outcomes.

## 2. Materials and Methods

### 2.1. Study Design and Participants

This cross-sectional study was conducted between April and June 2025 and enrolled 55 individuals with intellectual disabilities residing in two designated support facilities in Kumamoto, Japan. The participating facilities had 30 and 50 residents, respectively. Inclusion criteria were: (i) age ≥ 18 years; (ii) residence at a participating facility during the study period; (iii) a documented diagnosis of intellectual disability in physician-verified medical records; and (iv) oral intake feasible with or without assistance (self- or proxy-assisted assessment permitted). Individuals who were entirely non-communicative, receiving inpatient medical care, or reliant on enteral nutrition were excluded. The type and degree of intellectual disability were obtained from physician-verified medical records; no de novo classification was performed by the authors. The study was carried out in accordance with the ethical principles of the Declaration of Helsinki and was approved by the Ethics Committee of Kyushu Dental University (Approval No. 23–30; Approval Date: 13 October 2023). Written informed consent was obtained from the parents or legally authorized representatives of all participants. One family declined participation; this individual was not enrolled. During body composition assessment using bioelectrical impedance analysis (BIA), 6 participants moved despite instructions, resulting in measurement failure and exclusion because baseline BIA data were unavailable. Accordingly, 48 participants were included in the analyses.

### 2.2. Data Collection

Sociodemographic information, including age, length of stay in the facility, and gender, was obtained from facility records. The degree of intellectual disability was determined based on the “Ryoiku Techo” (Rehabilitation Handbook). According to the classification criteria in Kumamoto Prefecture, disability levels were divided into four categories: A1, A2, B1, and B2. Although these classifications are not based solely on IQ (intelligence quotient), the general criteria are as follows: A1 (profound): IQ below 20, requiring full assistance in all aspects of life; A2 (severe): IQ below 35, requiring full assistance in daily living; B1 (moderate): IQ below 50, requiring assistance in daily living; B2 (mild): IQ below 70, generally able to live independently.

Blood samples were collected using standard procedures to assess serum levels of triglycerides, high-density lipoprotein cholesterol, low-density lipoprotein cholesterol, albumin, and glucose, as indicators of nutritional status.

### 2.3. Health-Related Quality of Life

HRQOL was assessed using the Japanese version of the EQ-5D-5L, which includes five dimensions of health status: mobility, self-care, usual activities, pain/discomfort, and anxiety/depression [16]. Each dimension is rated on five levels, ranging from no problems to extreme problems. The resulting health states were converted into a single summary index with Microsoft Excel using the Japanese value set. Use of the EQ-5D-5L was licensed through the EuroQol Research Foundation (Registration ID: 65898), and the scoring was conducted in accordance with EuroQol guidelines. Subjective health perception was assessed by asking participants to rate their own health on a visual analog scale ranging from 0 to 100, where 0 indicated “the worst imaginable health” and 100 indicated “the best imaginable health.”

Given that many participants had limited verbal communication abilities or difficulty understanding abstract concepts, the EQ-5D-5L and EAT-10 were administered through structured interviews conducted by a trained dentist. All assessments were proxy-reported by facility staff members most familiar with each participant’s daily functioning; no self-reports were collected. Proxies were instructed to respond from the participant’s perspective, and all items were read verbatim from a standardized script. This approach follows prior research in populations with intellectual disabilities, where structured proxy interviews are commonly used to enhance response validity while respecting cognitive and communication limitations [17].

### 2.4. Nutritional Status

Nutritional status was assessed using the Mini Nutritional Assessment-Short Form (MNA-SF^®^), a validated screening instrument applicable to individuals with intellectual disabilities [18]. The MNA-SF^®^ comprises six domains: reduction in food intake, recent unintentional weight loss, physical mobility, experiences of acute illness or psychological stress, presence of neuropsychological conditions, and body mass index (BMI). Total scores range from 0 to 14, with lower values reflecting a greater risk of malnutrition. Trained facility staff conducted the evaluations. Anthropometric measurements, height and weight, were taken to compute BMI. Handgrip strength was evaluated using a digital hand dynamometer (YD, Tsutsumi Co., Tokyo, Japan) under standardized testing conditions.

### 2.5. Activity of Daily Living

Two widely used scales were employed to evaluate participants’ activities of daily living: the Barthel Index and the Functional Independence Measure (FIM). The Barthel Index is composed of 10 items related to basic self-care and mobility (e.g., feeding, dressing, toileting, bathing, transferring, ambulation, stair use, and continence). Each item is rated based on the degree of independence, with a cumulative maximum score of 100, where higher scores reflect greater autonomy [19]. The FIM provides a more detailed view of functional ability, encompassing 18 items categorized into motor and cognitive subdomains [20]. The motor domain includes self-care activities, bladder and bowel control, transfers, and locomotion. The cognitive domain evaluates communication abilities and social cognition, including problem-solving and memory. Each item is scored on a 7-point scale, yielding a total possible score of 126. Facility staff familiar with participants’ daily routines performed all assessments.

### 2.6. Oral Function

Swallowing function was assessed using the Eating Assessment Tool-10 (EAT-10), which consists of 10 items related to swallowing and weight loss [21]. Each item is rated from 0 to 4, with higher total scores indicating greater swallowing dysfunction. Based on the established cutoff score of 3 for dysphagia risk, participants were categorized into two groups: those scoring below 3 and those scoring 3 or higher. Because many participants had limited verbal communication abilities, the EAT-10 was administered through structured interviews with facility staff who were familiar with the participants’ daily eating behaviors. These staff members served as proxy respondents and were instructed to answer from the perspective of the participant rather than their own judgment. All interviews were conducted by a single trained dentist using a standardized protocol, including consistent phrasing and clarifying examples, to minimize interviewer and proxy bias. The Cronbach’s alpha for EAT-10 in this study was 0.858, indicating good internal consistency.

Oral bacterial load was measured using an oral bacteria counter (NP-BCM01-A, Panasonic, Tokyo, Japan) [22]. Sterile swabs were mounted on a pressure-controlled collection device and used to collect samples from the dorsum of the tongue. The swabs were then placed under the tongue for 10 s to absorb saliva, immersed in a sample preparation solution, and analyzed for impedance changes between electrodes to estimate bacterial counts. Bacterial counts were log-transformed for statistical analysis. The degree of tongue coating was evaluated using the Tongue Coating Index (TCI), which divides the tongue dorsum into nine segments—three each from the anterior, middle, and posterior regions [23]. Each segment was visually examined under adequate lighting and assigned a score from 0 to 2, where 0 indicated no visible coating, 1 represented a thin layer, and 2 denoted a thick coating. The cumulative TCI score, ranging from 0 to 18, was used to quantify overall coating. All assessments were conducted by a calibrated dentist trained in oral inspection procedures.

To assess tongue strength, we employed the TPM-01 device (JMS Co., Tokyo, Japan), which uses a balloon-type pressure sensor positioned on the tongue [24]. Participants were instructed to compress the sensor with maximal effort, and the highest value from three trials was used in the analysis. The device digitally recorded pressure data and provided real-time readouts.

The number of present teeth was determined by a clinical oral examination conducted by a dentist.

### 2.7. Body Composition

Body composition was measured using a multi-frequency bioelectrical impedance analyzer (seca mBCA 515, seca GmbH, Hamburg, Germany), which has been validated in populations with functional disabilities [25]. Participants lay supine during the procedure, and impedance values were obtained through eight surface electrodes placed on the hands and feet, spaced approximately 5 cm apart. The device calculated segmental impedance across the limbs and trunk and derived three key indices: fat mass (kg/m^2^), fat-free mass (kg/m^2^), and skeletal muscle mass (kg/m^2^), using proprietary algorithms that integrate anthropometric and demographic data (height, weight, sex, and age).

### 2.8. Statistical Analysis

Continuous variables were expressed as medians with minimum and maximum values, and categorical variables were presented as counts and percentages. Between-group comparisons between participants with EAT-10 scores ≤ 2 vs. ≥3 were conducted using the Mann–Whitney U test for continuous variables and the chi-square test for categorical variables. To examine factors associated with HRQOL, a generalized linear model (GLM) was constructed with EQ-5D-5L index scores as the dependent variable. Because HRQOL scores were not normally distributed and exhibited ceiling effects, they were summarized using the median and interquartile range (IQR). To appropriately model this non-normal distribution, a GLM with gamma distribution and identity link function was employed. Independent variables included the dichotomized EAT-10 group (based on the cutoff score of 3), age, sex, number of teeth, Barthel Index, handgrip strength, and MNA-SF^®^ score. Model fit was evaluated using the likelihood ratio test. To examine the robustness of the association between swallowing function and HRQOL observed in the main GLM using dichotomized EAT-10 scores, a sensitivity analysis was conducted by entering EAT-10 as a continuous variable. In all models, multicollinearity among independent variables was assessed using Spearman’s rank correlation coefficients. No strong correlations were observed among predictors (all |r| < 0.7), indicating that multicollinearity was within acceptable limits. Because all assessments were proxy-reported, respondent type was not modeled as a covariate. We additionally report post hoc power for the primary model (F-test based on observed R^2^) and for the adjusted incremental effect of EAT-10 (partial R^2^; F-test for R^2^ increase). All statistical analyses were performed using SPSS^®^ Statistics version 28.0 (IBM Corp., Tokyo, Japan). A *p*-value of less than 0.05 was considered statistically significant.

## 3. Results

Of the 54 individuals who provided consent, 6 did not enter the analysis set (BIA measurement failure due to movement: *n* = 6). Thus, 48 participants with intellectual disabilities were included in the statistical analysis, comprising 28 males (58.3%) and 20 females (41.7%). All assessments were proxy-reported. The median HRQOL score assessed using the EQ-5D-5L was 0.54 (range: 0.03–1.00; interquartile range (IQR): 0.39–0.66). Table 1 shows the comparison of participant characteristics based on swallowing function status, categorized by EAT-10 scores. The group with EAT-10 ≤ 2 had significantly higher scores in EQ-5D-5L [median 0.70 (0.30–1.00, IQR: 0.46–0.76) vs. 0.40 (0.00–0.60, IQR: 0.25–0.55), *p* < 0.001], perceived health [80 (40–100) vs. 70 (40–100), *p* = 0.026], FIM [76 (4–116) vs. 33 (0–99), *p* = 0.004], Barthel Index [70 (30–100) vs. 40 (10–75), *p* < 0.001], number of teeth [25 (6–31) vs. 21.5 (0–31), *p* = 0.035], grip strength [16.0 (5.8–29.0) kg vs. 10.9 (8.3–20.3) kg, *p* = 0.011], and fat mass [7.7 (0.5–14.5) kg/m^2^ vs. 4.9 (3.2–7.8) kg/m^2^, *p* = 0.028] compared to the group with EAT-10 ≥ 3.

To identify factors associated with HRQOL, a GLM was constructed using the EQ-5D-5L score as the dependent variable. Spearman’s rank correlation coefficients were calculated to evaluate multicollinearity among candidate independent variables (see Appendix A). Spearman’s rank correlation coefficients revealed multicollinearity among the following variable pairs: FIM and Barthel Index (*r* = 0.909), BMI and fat mass (*r* = 0.892), BMI and MNA-SF (*r* = 0.751), and fat mass and MNA-SF (*r* = 0.723). Accordingly, the following variables were selected for the GLM: dichotomized EAT-10 group (based on the cutoff score of 3), age, sex, number of teeth, Barthel Index, grip strength, and MNA-SF^®^ score (Table 2). In the GLM, an EAT-10 score ≤ 2 was positively associated with EQ-5D-5L score [B ± SE = 0.37 ± 0.08; 95% Wald confidence interval: 0.22 to 0.52; *p* < 0.001]. To assess the robustness of this finding, a sensitivity analysis was conducted using EAT-10 as a continuous variable (see Appendix A). The result showed a similar negative association with EQ-5D-5L score [B ± SE = −0.05 ± 0.01; 95% Wald confidence interval: −0.06 to −0.03; *p* < 0.001], indicating that higher EAT-10 scores (greater risk of dysphagia) were consistently associated with lower HRQOL. Although the dichotomized EAT-10 group (≤2 vs. ≥3) showed a positive association with EQ-5D-5L scores and the EAT-10 total score (continuous) showed a negative association, both results consistently indicate that better swallowing function is associated with higher QOL.

Model-level observed power was 1 − β = 0.96; for EAT-10 (adjusted), partial R^2^ = 0.368, corresponding to f^2^ =0.582.

## 4. Discussion

This study demonstrated a significant association between swallowing dysfunction and lower HRQOL among individuals with intellectual disabilities. Participants identified as being at higher risk of dysphagia, based on EAT-10 scores had markedly lower EQ-5D-5L scores, indicating compromised HRQOL. Notably, this relationship remained consistent even when the EAT-10 score was treated as a continuous variable in sensitivity analyses, supporting the robustness of the findings (Appendix A). These results provide clear evidence that swallowing difficulties contributes to reduced HRQOL in this population, a topic that has received limited empirical attention to date. The present findings underscore the importance of routine screening and early intervention for swallowing difficulties in individuals with intellectual disabilities, not only to prevent medical complications such as aspiration pneumonia, but also to improve overall well-being and daily functioning. Although most prior studies have focused primarily on physical health outcomes, the present study highlights the broader psychosocial impact of swallowing difficulties on subjective well-being. This is in line with findings from a study on older adults with functional decline, in which swallowing ability assessed by the Functional Oral Intake Scale was strongly correlated with QOL measured by the short version of the Quality of Life Questionnaire for Dementia [26]. Given that individuals with intellectual disabilities may have limited ability to articulate discomfort or report symptoms, the use of structured tools such as EAT-10 and EQ-5D-5L offers a valuable framework for identifying individuals at risk and guiding multidisciplinary interventions. By addressing swallowing difficulties proactively, care providers may contribute not only to the prevention of life-threatening complications but also to the enhancement of QOL in this vulnerable population.

In this study, swallowing dysfunction assessed by the EAT-10 was significantly associated with malnutrition as evaluated by the MNA-SF^®^. This finding is consistent with previous reports indicating that individuals with intellectual disabilities and neurological impairments are particularly vulnerable to dehydration and undernutrition when swallowing difficulties are present [5]. Feeding difficulties, including problems in all three phases of swallowing, have been well documented in children with Down syndrome, with greater difficulty observed for solids than liquids [27]. Among older adults in general, numerous studies have shown that reduced swallowing function is a significant risk factor for undernutrition [24,28,29]. In the present study, the participants with EAT-10 scores ≥ 3 were significantly older than those with lower scores, as shown in Table 1. This suggests that even in adults with intellectual disabilities without overt neurological disorders such as cerebral palsy, aging may contribute to the decline of swallowing function and, consequently, increase the risk of malnutrition. Similarly, swallowing dysfunction was also significantly associated with reduced grip strength. Prior studies in older adults have reported associations between oral function, including tongue and lip movements, and physical fitness indicators such as handgrip strength and walking speed [30,31]. Our findings suggest that a similar relationship may exist among individuals with intellectual disabilities. Grip strength, a well-established marker of general muscle function and nutritional status, may reflect systemic muscular decline associated with dysphagia in this population as well.

In this study, participants in the EAT-10 ≤ 2 group had significantly lower fat mass compared to those in the EAT-10 ≥ 3 group. Reference values for fat mass are typically 3–6 kg/m^2^ for men and 5–9 kg/m^2^ for women [32]. In our sample, the overall fat mass was within these ranges for both sexes—5.2 (0.5–12.8) kg/m^2^ for men and 7.8 (3.2–14.5) kg/m^2^ for women. However, in the EAT-10 ≥ 3 group, both male and female participants showed a trend toward lower fat mass, with median values of 4.9 (3.2–7.7) kg/m^2^ in men and 5.4 (3.2–7.8) kg/m^2^ in women. Adipose tissue is an endocrine organ (e.g., leptin, adiponectin) that also modulates sex-steroid and glucocorticoid metabolism; insufficient fat mass can disturb physiological regulation [33]. In our cohort, those with swallowing dysfunction showed lower fat mass despite normal lipid/glucose profiles, suggesting adipose loss reflects atrophy from reduced intake or systemic catabolism rather than macronutrient deficiency. In the nutritional management of individuals with intellectual disabilities, it is essential to monitor not only skeletal muscle mass but also the preservation of adequate fat mass to ensure metabolic and endocrine homeostasis.

The number of remaining teeth has been associated with HRQOL in older adults [34,35]. Although a few studies in individuals with intellectual disabilities have reported improvements in oral HRQOL following dental treatment [36], to our knowledge, there are no studies specifically examining the relationship between the number of remaining teeth and general HRQOL in this population. Therefore, further research is warranted to clarify this association. In addition, previous studies have demonstrated that tooth loss is related to a decline in swallowing function [29,37], suggesting that the number of remaining teeth may influence overall QOL indirectly through its impact on swallowing ability.

The use of the EQ-5D-5L to assess HRQOL was a strength of this study. As a brief, validated, and widely used generic measure, it was suitable for individuals with intellectual disabilities due to its low cognitive burden and ease of administration when supported by caregivers or staff [38]. Moreover, the EQ-5D-5L enabled us to describe and statistically analyze the relationship between swallowing function and HRQOL scores. The utility-based scoring system is also advantageous for future health economic evaluations.

Although the EAT-10 was originally developed as a self-administered tool to capture subjective swallowing difficulties, its use via proxy respondents in this study warrants consideration. Due to cognitive and communicative limitations, most participants were unable to complete the questionnaire independently. Therefore, proxy responses were provided by facility staff members who were familiar with the participants’ daily eating behaviors and health conditions. To enhance standardization and reduce potential bias, a trained dentist conducted all interviews using a structured protocol and clarifying examples. While proxy administration of the EAT-10 is not widely validated, recent studies have reported its feasibility in cognitively impaired populations when proxies are well-informed caregivers [39]. In the present study, the EAT-10 showed high internal consistency (Cronbach’s alpha = 0.858), supporting its reliability in this context. These findings suggest that, despite its limitations, the EAT-10 can serve as a feasible screening tool for swallowing dysfunction among individuals with intellectual disabilities when administered with appropriate methodological safeguards.

Using EAT-10 and the EQ-5D-5L, we demonstrated that a higher risk of dysphagia is independently associated with lower HRQOL scores. Both instruments were administered via proxy respondents (facility staff) due to participants’ cognitive limitations. Prior validation studies have shown that proxy-based assessments of EQ-5D-5L are appropriate and reliable in populations with intellectual disabilities. In our study, there was no indication of systematic response bias for either the EQ-5D-5L or the EAT-10. These findings emphasize the importance of including swallowing assessments in multidisciplinary care and highlight a potential pathway for improving the QOL in this vulnerable population.

Our findings suggest that limited cognition and communication may hinder the implementation of dysphagia recommendations. In practice, care plans should therefore be individualized and supported by clear, simplified instructions and appropriate caregiver education. How best to optimize communication and comprehension in this context remains an important topic for future research.

From a population-health perspective, our findings support routine dysphagia screening as part of regular health checks in residential facilities. In addition, public-health implication for service planning, we propose (i) integrated oral health–nutrition–swallowing care pathways (including timely dental/oral assessments and mealtime management) and (ii) caregiver training in safe feeding and recognition of early warning signs. Because proxy reports are common in this setting, system-level protocols are needed to ensure reliability and to trigger referral when risk is identified.

Several limitations of this study should be acknowledged. First, the cross-sectional design limits causal interpretation between swallowing function and HRQOL. Longitudinal studies are warranted to clarify the directionality of these associations. Second, the relatively small sample size, restricted to two facilities in a single geographic area, may limit the generalizability of the findings to broader populations of individuals with intellectual disabilities. Third, although swallowing function was assessed using the EAT-10, which is a validated questionnaire, it was administered through interviews conducted by a dentist rather than self-completed by participants. While this approach helped ensure response accuracy in a population with communication challenges, it may still be subject to proxy response bias. Fourth, although we controlled several relevant covariates using GLM, the potential influence of unmeasured factors, such as cognitive function, emotional well-being, or medication use, cannot be ruled out. Cognitive function may influence both the ability to communicate symptoms and the perceived QOL. Likewise, the use of medications such as antipsychotics or anticonvulsants, which are prevalent in this population, could affect swallowing function or mood. While these variables were not systematically assessed in the present study, future research should include them to better isolate the independent contribution of swallowing dysfunction to HRQOL. Fifth, all outcomes were proxy-reported; thus, proxy-report bias is possible, although we sought to mitigate this by using standardized interviews and proxies most familiar with each participant. Sixth, we did not assess xerostomia or salivary flow; therefore, residual confounding by salivary function cannot be excluded. Future studies should incorporate validated measures of oral dryness and salivary secretion to better delineate their relationship with dysphagia and HRQOL. Seventh, we did not have standardized data on cognitive level or psychotropic medication use (e.g., antipsychotics, antiepileptics), so these potential confounders could not be adjusted for and residual confounding cannot be excluded.

## 5. Conclusions

This study provides empirical evidence that swallowing dysfunction is significantly associated with lower HRQOL among adults with intellectual disabilities residing in residential care settings. Participants with higher EAT-10 scores, indicating greater dysphagia risk, consistently reported poorer HRQOL as measured by the EQ-5D-5L. These findings underscore the need to recognize dysphagia not only as a clinical concern, but also as a factor that influences broader aspects of well-being in this population.

Given the challenges in self-reporting among individuals with intellectual disabilities, structured assessments such as the EAT-10 and EQ-5D-5L can serve as practical tools for identifying at-risk individuals and informing multidisciplinary care strategies. Integrating routine swallowing dysfunction screening into care practices may help improve not only physical health outcomes but also overall QOL.

## Figures and Tables

**Table 1 ijerph-22-01470-t001:** Comparison of participant characteristics by swallowing function status based on EAT-10 scores.

	EAT-10		*p*-Value
	≤2 (*n* = 31)	≥3 (*n* = 17)	
Age (y)	57 (36–90)	69 (47–85)	0.031 ^†^
Gender			
Male	16 (51.6%)	12 (70.6%)	0.202 ^‡^
Female	15 (48.4%)	5 (29.4%)	
EQ-5D-5L	0.7 (0.3–1)	0.4 (0–0.6)	<0.001 ^†^
Perceived health	80 (40–100)	70 (40–100)	0.026 ^†^
Institutionalizing period (y)	24 (2–37)	30 (10–37)	0.489 ^†^
Disability level			
A1	15 (48.4%)	10 (58.8%)	0.779 ^‡^
A2	12 (38.7%)	5 (29.4%)	
B1	3 (9.7%)	2 (11.8%)	
B2	1 (3.2%)	0 (0%)	
Activity of daily living			
FIM	76 (4–116)	33 (0–99)	0.004 ^†^
Barthel index	70 (30–100)	40 (10–75)	<0.001 ^†^
Oral condition			
Bacterial counts (log10)	7.3 (6.5–7.9)	7.5 (6.5–8.0)	0.064 ^†^
Number of teeth	25 (6–31)	21.5 (0–31)	0.035 ^†^
Tongue pressure (kPa)	13.4 (1–43)	7.0 (2–29)	0.306 ^†^
Tongue Coating Index	0 (0–18)	0 (0–15)	0.116 ^†^
Body composition			
Skeletal muscle mass (kg/m^2^)	14 (8.1–23)	12 (4.8–41.6)	0.065 ^†^
Fat-free mass (kg/m^2^)	15.1 (12–17)	14.8 (12.4–16.9)	0.518 ^†^
Fat mass (kg/m^2^)	7.7 (0.5–14.5)	4.9 (3.2–7.8)	0.028 ^†^
BMI (kg/m^2^)	22.0 (14.4–28.7)	19.6 (16.0–25.0)	0.065 ^†^
MNA-SF^®^	12 (6–14)	11 (9–13)	0.009 ^†^
Grip strength (kg)	16.0 (5.8–29.0)	10.9 (8.3–20.3)	0.011 ^†^
Serum concentration			
Triglyceride (mg/dL)	70 (27–184)	47 (0–153)	0.093 ^†^
HDL cholesterol (mg/dL)	69 (31–111)	70 (46–101)	0.906 ^†^
LDL cholesterol (mg/dL)	109 (60–151)	100 (45–166)	0.327 ^†^
Albumin (g/dL)	4.3 (3.8–5.4)	4.3 (3.7–4.8)	0.522 ^†^
Glucose (mg/dL)	88 (73–120)	85 (74–123)	0.305 ^†^

^†^: Mann–Whitney U test, ^‡^: chi-squared test. FIM: Functional Independence Measure, BMI: Body Mass Index, MNA-SF: Mini Nutritional Assessment-Short Form, HDL: High-Density Lipoprotein, LDL: Low-Density Lipoprotein.

**Table 2 ijerph-22-01470-t002:** Generalized linear model analysis of factors associated with EQ-5D-5L.

Variables	B ± SE	95% Wald CI ^a^	*p*-Value
EAT-10 ≤ 2	0.37 ± 0.08	0.22–0.52	<0.001
≥3	Reference		
Age	−0.01 ± 0	−0.01–−0.01	<0.001
Gender Female	−0.21 ± 0.07	−0.35–−0.08	0.001
Male	Reference		
Number of teeth	−0.02 ± 0	−0.09–0	0.077
Barthel index	0 ± 0	0–0	0.735
Handgrip	0.01 ± 0.01	0–0.02	0.077
MNA-SF^®^	−0.05 ± 0.02	−0.09-−0.01	0.018

^a^: confidence interval. The generalized linear model significantly improved model fit over the intercept-only model (likelihood ratio χ^2^(7) = 26.0, *p* < 0.001). R^2^ = 0.467, adjusted R^2^ = 0.307; variable-level contribution for EAT-10: partial η^2^ = 0.073.

## Data Availability

The data that support the findings of this study are available on request from the corresponding author. The data are not publicly available due to privacy or ethical restrictions.

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
