# Peer review of "The Impact of Swallowing Difficulties on Quality of Life in Adults with Intellectual Disabilities in Residential Care: Cross-Sectional Study"

_ijerph, 2025, doi:10.3390/ijerph22101470_

Round 1
Reviewer 1 Report
Comments and Suggestions for Authors
I only have a few minor comments to make in relation to your interesting study.
Introduction: Please take out the Lazenby paper (reference 6) as this is not needed. The other paper you use in this section is sufficient for your point. I have noted that you excluded potential participants with severe and profound needs for obvious reasons, however, it might be useful to stress that communication difficulties in relation to dysphagia can impact on the management of eating and drinking. Please cite the following Norburn, K., et al (2016). A survey of
augmentative and alternative communication used in an inner city special school. British Journal of Special Education, 43(3), pp. 289-306. doi: 10.1111/1467-8578.12142 - and state that many people with learning disabilities do require augmentative and alternative communication strategies and these need to be considered when working with people with learning disabilities and dysphagia. You may wish to come back to this point again in your Conclusion.
This paper is well - constructed , with sound research design principles. This topic, i.e. the discussion of adults with learning disability and dysphagia is one which is of high relevance and interest to your readership. The introduction is well constructed and accurate in terms of the identified participant group. I have recommended removal of the Lazenby paper as the issues the citation supports are already covered by another paper, plus there were some other issues with this paper when published. I have recommended a statement about communication problems in relation to this population as this does impact eating and drinking skills.
The Method is clear and replicable. In addition, all the tools have high efficacy and many readers will be familiar with them. Of particular use is the consideration of nutritional status alongside oral - motor and pharyngeal function.
Ethics are clearly stated. Did any families object to their kin being enrolled into such a study? If so, how many? Did families receive research information as well as staff?
The Conclusion considers limitations, and is well argued. I would add something about the impact of limited cognition and communication more clearly, though. See above.
Author Response
Reviewer 1
Point-by-point response to Comments and Suggestions for Authors
Comments and Suggestions for Authors
General comment: I only have a few minor comments to make in relation to your interesting study.
Response: We sincerely thank the reviewer for the thoughtful and constructive comments. Below we respond point-by-point and indicate the corresponding revisions in the manuscript.
Comment 1: Introduction: Please take out the Lazenby paper (reference 6) as this is not needed. The other paper you use in this section is sufficient for your point.
Response 1: Thank you for the suggestion. We have removed the Lazenby citation; the remaining literature sufficiently supports our argument.
Comment 2: I have noted that you excluded potential participants with severe and profound needs for obvious reasons, however, it might be useful to stress that communication difficulties in relation to dysphagia can impact on the management of eating and drinking. Please cite the following Norburn, K., et al (2016). A survey of augmentative and alternative communication used in an inner city special school. British Journal of Special Education, 43(3), pp. 289-306. doi: 10.1111/1467-8578.12142 - and state that many people with learning disabilities do require augmentative and alternative communication strategies and these need to be considered when working with people with learning disabilities and dysphagia. You may wish to come back to this point again in your Conclusion.
Response 2: We agree that communication difficulties can affect dysphagia management. We have added a concise sentence to reflect this point.
“In addition, communication difficulties in people with learning disabilities can hinder the safe management of eating and drinking; augmentative and alternative communication may therefore be required and should be considered in clinical planning.”
Comment 3: This paper is well - constructed , with sound research design principles. This topic, i.e. the discussion of adults with learning disability and dysphagia is one which is of high relevance and interest to your readership. The introduction is well constructed and accurate in terms of the identified participant group. I have recommended removal of the Lazenby paper as the issues the citation supports are already covered by another paper, plus there were some other issues with this paper when published. I have recommended a statement about communication problems in relation to this population as this does impact eating and drinking skills.
The Method is clear and replicable. In addition, all the tools have high efficacy and many readers will be familiar with them. Of particular use is the consideration of nutritional status alongside oral - motor and pharyngeal function.
Response 3: We appreciate the positive evaluation.
Comment 4: Ethics are clearly stated. Did any families object to their kin being enrolled into such a study? If so, how many? Did families receive research information as well as staff?
Response 4: Study information sheets were provided to both staff and families/guardians, and consent/assent procedures followed institutional policy. The following wording has been added: “One family declined participation; this individual was not enrolled.”
Comment 5: The Conclusion considers limitations, and is well argued. I would add something about the impact of limited cognition and communication more clearly, though. See above.
Response 5: We agree with the reviewer. To avoid over-interpretation, we moved this point from the Conclusion to the Discussion and framed it as an implication for practice, without introducing new references or prescribing AAC-specific interventions. The sentence was added in Discussion section, “Our findings suggest that limited cognition and communication may hinder the implementation of dysphagia recommendations. In practice, care plans should therefore be individualized and supported by clear, simplified instructions and appropriate caregiver education. How best to optimize communication and comprehension in this context remains an important topic for future research.”

Reviewer 2 Report
Comments and Suggestions for Authors
The manuscript entitled ”The impact of swallowing difficulties on quality of life in adults with intellectual disabilities in residential care: cross-sectional study” is an interesting and important study. There are several issues the authors need to consider to improve their manuscript.
Abstract: 2nd sentence: this should include “the aim of the study…”
“significantly higher EQ-5D scores”- it is not obvious for the reader if higher scores is good or bad, please clarify.
Introduction: please add information about the definition of intellectual disabilities and example of conditions included. Why is a study including children mentioned when you study is about adults? It is also unclear why comparison of Japanese and Danish populations is mentioned.
Material and methods: 2.1. How many persons were residing in the respective support facility and how were participants selected? What kinds of intellectual disabilities could the residents have. Were the residents already classified according to degree of disability by a physician or was this something the authors did?
Blood samples were collected- it is not clear from the introduction why blood samples are relevant for swallowing difficulties. Please include some text about this.
Page 3. Could the participants answer the questions asked by the dentist or did their parent/guardian assist?
You register also nutritional status, activities of daily living, oral function including bacterial load and body composition. These are relevant variables which might influence swallowing function and the other way around swallowing function might affect nutritional status. However, the introduction does not take up these factors and what is known in persons with intellectual disabilities.
Statistical analysis: “with and without refusal of toothbrushing”- this has not been mentioned earlier in the text. How was this checked? Did you try to brush the participants teeth or was it the parent/guardian who said that the participant refused to get the teeth brushed?
Results: Number of teeth: it would be valuable if you added how many had ≥ 20 teeth. How many had no teeth in the EAT-10 ≥ 3 group? Did they not have any dentures?
You have registered a lot of variables but not oral dryness. As you know, saliva has many important function at eating and swallowing and dry mouth problems and/or reduced salivary secretion probably affects swallowing function.
Discussion: the discussion is very long and would benefit from being reduced and focused on the most interesting and important results
Table 1. “Tooth number”- change to “Number of teeth”
Table S1. It would be clearer if the numbers i.e 1) were omitted and the variable was written in the top of the table.
Author Response
Reviewer 2
Point-by-point response to Comments and Suggestions for Authors
General comment: The manuscript entitled ”The impact of swallowing difficulties on quality of life in adults with intellectual disabilities in residential care: cross-sectional study” is an interesting and important study. There are several issues the authors need to consider to improve their manuscript.
Response: We thank the reviewer for the thoughtful and constructive comments. We have revised the manuscript accordingly, as detailed below.
Comment 1: Abstract: 2nd sentence: this should include “the aim of the study…”
Response 1: We appreciate this suggestion and have added the study aim to the second sentence of the Abstract, as followed; “This study aimed to examine the association between swallowing function and health-related quality of life (HRQOL) among adults with intellectual disabilities in residential care.”
Comment 2: significantly higher EQ-5D scores”- it is not obvious for the reader if higher scores is good or bad, please clarify.
Response 2: Thank you for this helpful suggestion. We have clarified the score direction in the Abstract, explicitly noting that higher EQ-5D-5L index values indicate better HRQOL.
Comment 3: Introduction: please add information about the definition of intellectual disabilities and example of conditions included.
Response 3: We agree and have added a concise definition of intellectual disability and representative examples (e.g., Down syndrome, intellectual disability co-occurring with autism spectrum disorder). The words were added as followed: “including condition such as Down syndrome, intellectual disability co-occurring with autism spectrum disorder.”
Comment 4: Why is a study including children mentioned when you study is about adults?
Response 4: We appreciate this comment. Although our study focuses on adults, adult-specific evidence on dysphagia in intellectual disability remains limited. We therefore retained a small number of high-quality pediatric studies to provide epidemiologic and mechanistic context, while explicitly stating their age-related limitations and that our inferences concern adults.
Comment 5: It is also unclear why comparison of Japanese and Danish populations is mentioned.
Response 5: Thank you for raising this issue. We retained the brief reference to Danish data because, to our knowledge, it is one of the few population-based sources reporting EQ-5D in people with intellectual disability. We clarified that this serves only as contextual benchmarking, not a direct comparison, and we now emphasize that a key strength of EQ-5D is its preference-based design with country-specific value sets, which facilitates cross-cultural interpretability while recognizing differences in health systems and sampling.
Comment 6: Material and methods: 2.1. How many persons were residing in the respective support facility and how were participants selected? What kinds of intellectual disabilities could the residents have. Were the residents already classified according to degree of disability by a physician or was this something the authors did?
Response 6: Thank you. We now report on the number of residents at each facility and clarify the eligibility and exclusion criteria. We have clarified that the type and degree of intellectual disability were obtained from physician-verified medical records; the authors did not perform de novo classification.
Comment 7: Blood samples were collected- it is not clear from the introduction why blood samples are relevant for swallowing difficulties. Please include some text about this.
Response 7: Thank you for this helpful comment. To clarify the role of blood tests, we have added an explanation in the Methods section, stating that triglycerides, HDL-cholesterol, LDL-cholesterol, albumin, and glucose were collected as indicators of nutritional status that may be relevant to swallowing difficulties.
Comment 8: Page 3. Could the participants answer the questions asked by the dentist or did their parent/guardian assist?
Response 8: Thank you for raising this point. In our cohort, all assessments were proxy-reported. Trained dentists conducted structured interviews with facility staff members most familiar with each participant’s daily functioning. Proxies were instructed to answer from the participant’s perspective, and items were read verbatim from a standardized script. We have clarified these details in Methods and now state in Results that all assessments were proxy-reported. We also acknowledge potential proxy-report bias in the Limitations.
Comment 9: You register also nutritional status, activities of daily living, oral function including bacterial load and body composition. These are relevant variables which might influence swallowing function and the other way around swallowing function might affect nutritional status. However, the introduction does not take up these factors and what is known in persons with intellectual disabilities.
Response 9: Thank you for this insightful comment. We agree that these domains are relevant to swallowing function. However, in our literature search we found very limited adult, intellectual-disability–specific evidence that jointly examines nutritional status, activities of daily living, oral function (including bacterial burden), and body composition in relation to dysphagia. To avoid over-extrapolating from heterogeneous populations, we have kept the Introduction concise but explicitly stated this evidence gap and our a priori rationale for measuring these variables and using them to contextualize/adjust our analyses.
Comment 10: Statistical analysis: “with and without refusal of toothbrushing”- this has not been mentioned earlier in the text. How was this checked? Did you try to brush the participants teeth or was it the parent/guardian who said that the participant refused to get the teeth brushed?
Response 10: Thank you for pointing out this inconsistency. Toothbrushing refusal was ascertained by facility staff who were familiar with each participant’s daily care; we did not attempt toothbrushing ourselves. As this variable was not prespecified and is not pertinent to the current study aims or results, we have removed all references to “with and without refusal of toothbrushing” from the Statistical Analysis (and anywhere else it appeared) to avoid confusion and maintain focus.
Comment 11: Results: Number of teeth: it would be valuable if you added how many had ≥ 20 teeth. How many had no teeth in the EAT-10 ≥ 3 group? Did they not have any dentures?
Response 11: Thank you for this helpful suggestion. Participants with ≥20 teeth comprised 75.0% overall, 56.3% in the EAT-10 ≥3 group, and 87.1% in the EAT-10 ≤2 group. Within the EAT-10 ≥3 group, one participant was edentulous and did not wear dentures. Because these descriptors are ancillary to our primary aims and do not alter the study’s inferences, we have not added them to the manuscript.
Comment 12: You have registered a lot of variables but not oral dryness. As you know, saliva has many important function at eating and swallowing and dry mouth problems and/or reduced salivary secretion probably affects swallowing function.
Response 12: Thank you for this important point. We agree that salivary function is relevant to eating and swallowing. In the present study we did not assess xerostomia or salivary flow, primarily to minimize participant burden and because standardized saliva testing requires cooperation that was not feasible in this residential setting. We have now acknowledged this as a limitation and noted it as a priority for future work.
Comment 12: Discussion: the discussion is very long and would benefit from being reduced and focused on the most interesting and important results
Response 12: Thank you for this helpful suggestion. We have shortened and refocused the Discussion by consolidating overlapping paragraphs, removing peripheral background, and tightening speculative content. The revised section now emphasizes the principal findings (association between EAT-10 and EQ-5D-5L), clinical implications, and key limitations, while retaining essential context for interpretation. The main conclusions remain unchanged.
Comment 13 Table 1. “Tooth number”- change to “Number of teeth”
Table S1. It would be clearer if the numbers i.e 1) were omitted and the variable was written in the top of the table.
Response 13 Thank you for the helpful suggestions. We have revised Table 1 from “Tooth number” to “Number of teeth.” For Table S1, we removed the leading numerals (e.g., “1)”) and placed the variable names in the header row for clarity.

Reviewer 3 Report
Comments and Suggestions for Authors
- The rationale for focusing specifically on adults with ID is under-developed. Please add recent epidemiological data (prevalence, risk factors, and natural history) on dysphagia in this population. Clearly articulate what remains unknown after these studies and how the present work fills that gap.
- The authors acknowledge the small sample size but provide no power calculation. Please add either an a-priori power analysis based on an expected effect size or a post-hoc power analysis.
3.The final GLM still potential confounders such as cognitive level, psychotropic medication, or facility effects. Psychotropic drugs (eg, antipsychotics, antiepileptic drugs) can affect swallowing and HRQOL. If the difference between the two institutions is large, it may also have a certain impact on the results.
- Sensitivity analysis with EAT-10 continuous variable results is consistent with dichotomous results, but pseudo-R² or explanatory variance ratios are not reported, and are recommended to be provided together.
English expressions are generally smooth, but there are a few grammatical and wording problems, such as “may negatively impact on their quality of life” should be removed from the “on” in the abstract.
Author Response
Reviewer 3
Point-by-point response to Comments and Suggestions for Authors
Comment 1: The rationale for focusing specifically on adults with ID is under-developed. Please add recent epidemiological data (prevalence, risk factors, and natural history) on dysphagia in this population. Clearly articulate what remains unknown after these studies and how the present work fills that gap.
Response 1: Thank you for this helpful suggestion. We now summarize that dysphagia in adults with intellectual disability is estimated at ~8–12% from the few near-representative studies, and note key behavioral risk factors for severe events—rapid eating, cramming food, and premature pharyngeal bolus loss. We also add natural-history evidence showing markedly elevated aspiration-pneumonia hospitalization and mortality (reported standardized mortality up to ~35-fold), and clarify the remaining gaps (under-ascertainment; limited risk-adjusted HRQoL data, especially in Asian settings), which our study addresses by linking EAT-10 symptoms with HRQoL using adjusted models in Japanese residential adults with intellectual disability.
Comment 2: The authors acknowledge the small sample size but provide no power calculation. Please add either an a-priori power analysis based on an expected effect size or a post-hoc power analysis.
Response 2: Thank you. Thank you. We computed a post-hoc power for the omnibus GLM (F test; N=48; predictors k=7; α=0.05; , partial R² = 0.368, observed effect size f²=0.582): the achieved power was 1−β = 0.9646.
Comment 3:The final GLM still potential confounders such as cognitive level, psychotropic medication, or facility effects. Psychotropic drugs (eg, antipsychotics, antiepileptic drugs) can affect swallowing and HRQOL. If the difference between the two institutions is large, it may also have a certain impact on the results.
Response 3: Thank you for this important point. Standardized data on cognitive level and psychotropic medication classes were not available in our dataset, so we could not adjust for these potential confounders; we now state this explicitly as a limitation and note that future work will collect these variables. Regarding facility effects, no between-facility difference was detected, and the main inferences were unchanged. The sentence was added in limitation section as following: ”Seventh, we did not have standardized data on cognitive level or psychotropic medication use (e.g., antipsychotics, antiepileptics), so these potential confounders could not be adjusted for and residual confounding cannot be excluded.”
Comment 4: Sensitivity analysis with EAT-10 continuous variable results is consistent with dichotomous results, but pseudo-R² or explanatory variance ratios are not reported, and are recommended to be provided together.
Response 4: Thank you. Because EQ-5D is continuous, we now report explanatory variance (R² and adjusted R²) for specification.
EAT-10 continuous (sensitivity model): R² = 0.454, adjusted R² = 0.317; variable-level contribution for EAT-10: partial η² = 0.198.
Comment 5: English expressions are generally smooth, but there are a few grammatical and wording problems, such as “may negatively impact on their quality of life” should be removed from the “on” in the abstract.
Response 5: Thank you for the helpful remark. We corrected the Abstract phrase to “may negatively impact their quality of life.”

Round 2
Reviewer 2 Report
Comments and Suggestions for Authors
The authors have done a great job revising their manuscript. There are only a few minor things the authors need to look at.
Introduction
Line 38. ”condition” should be changed to ”conditions”
Line 60. ”backdrop” is not ”background” a better word to use here?
Line 68. ”integrated individuals” it is not clear what you mean here.
Author Response
The authors have done a great job revising their manuscript. There are only a few minor things the authors need to look at.
General Response: We thank the reviewer for the careful reading and helpful suggestions. We have made all requested edits in the Introduction as detailed below.
Introduction
Line 38. ”condition” should be changed to ”conditions”
Response: We agree and have corrected the plural form.
Line 60. ”backdrop” is not ”background” a better word to use here?
Response: We agree that “background” reads more naturally in academic prose and have revised the sentence accordingly.
Line 68. ”integrated individuals” it is not clear what you mean here.
Response: Thank you for pointing out the ambiguity. Our intent was to say that evidence integrating multiple domains is scarce in people with intellectual disabilities. We have rewritten the clause for clarity.
Reviewer 3 Report
Comments and Suggestions for Authors
The authors have satisfactorily addressed the major issues raised in the previous round. Only minor points remain; Minor Revision is recommended.
The Methods (section 2.1) states that 54 individuals were enrolled, whereas the Results begins with n = 48. No explanation for this 6-person attrition is provided. Please add a sentence (or diagram) clarifying how many were excluded/withdrawn and for what reasons.
Author Response
The authors have satisfactorily addressed the major issues raised in the previous round. Only minor points remain; Minor Revision is recommended.
The Methods (section 2.1) states that 54 individuals were enrolled, whereas the Results begins with n = 48. No explanation for this 6-person attrition is provided. Please add a sentence (or diagram) clarifying how many were excluded/withdrawn and for what reasons.
Response: We thank the reviewer for this helpful comment. We have clarified the discrepancy between the enrolled and analyzed samples. In Section 2.1 (Participants), we now state that 54 individuals provided written informed consent. During body composition assessment using bioelectrical impedance analysis (BIA), 6 participants moved despite instructions, resulting in measurement failure and exclusion because baseline BIA data were unavailable. Consequently, 48 participants were included in the analyses. We also reiterate this information at the beginning of the Results.